# Evaluation of Neuroinflammatory Contribution to Neurodegeneration in LRRK2 *Drosophila* Models

**DOI:** 10.3390/biomedicines12071555

**Published:** 2024-07-12

**Authors:** Hoai Nam Nguyen, Grazia Galleri, Antonio Rassu, Cristina Ciampelli, Roberto Bernardoni, Manuela Galioto, Diego Albani, Claudia Crosio, Ciro Iaccarino

**Affiliations:** 1Department of Biomedical Sciences, University of Sassari, 07100 Sassari, Italygalleri@uniss.it (G.G.); ntnrassu@gmail.com (A.R.);; 2Department Pharmacy and Biotechnology, University of Bologna, 40126 Bologna, Italy; roberto.bernardoni@unibo.it; 3Department of Agricultural Sciences, University of Sassari, 07100 Sassari, Italy

**Keywords:** LRRK2, glial cells, levetiracetam, neuroinflammation, Parkinson’s disease

## Abstract

Pathological mutations in the *LRRK2* gene are the major genetic cause of Parkinson’s disease (PD). Although several animal models with either LRRK2 down- or over-expression have been developed, the physiological function of LRRK2 remains elusive. *LRRK2* is constitutively expressed in various tissues including neurons and glial cells, but importantly, it is expressed at low levels in dopaminergic neurons, further contributing to the cryptic function of *LRRK2*. Significant levels of LRRK2 protein and mRNA have been detected in peripheral blood mononuclear cells, lymph nodes, the spleen, and primary microglia, strongly suggesting the contribution of inflammatory cells to neuronal degeneration. In this research article, using *Drosophila* LRRK2 models, we were able to demonstrate a significant contribution of glial cells to the LRRK2 pathological phenotype. Furthermore, in *Drosophila*, neurodegeneration is associated with a significant and important increase in specific inflammatory peptides. Finally, levetiracetam, a compound widely used in human therapy to treat epilepsy, was able to rescue both neuronal degeneration and neuroinflammation.

## 1. Introduction

Pathological mutations in the Leucine-Rich Repeat Kinase 2 (*LRRK2*) gene are the most common genetic cause of Parkinson’s disease [1,2]. In addition, different mutations in the PARK8 locus have been identified as risk factors for PD. The penetrance of LRRK2 mutation is quite low, and it is not uncommon to find two siblings carrying the same pathological mutations who have completely different pathological conditions [3], underlining the importance of other genetic and/or environmental factors in PD onset. From a clinical perspective, patients carrying LRRK2 mutations are almost indistinguishable from idiopathic ones, while, at the neuropathological level, a prevalence of tau vs. synuclein aggregates in LRRK2 mutation carriers has been highlighted [4].

LRRK2 is a member of the ROCO family of proteins, which are characterised by a Ras-like GTPase domain, called Roc, followed by a COR (C-terminal of Roc) domain. LRRK2 contains several protein domains: armadillo repeats (ARM), ankyrin repeats (ANK), leucine-rich repeats (LRR), Roc, COR, kinase, and the WD40 domain [5]. To date, *LRRK2* pathological mutations are considered autosomal dominants, since no *LRRK2* nonsense mutations have been identified in association with PD. The pathological mutations are located around the central catalytic core of the protein, two mutations are found in the Roc domain, one in the COR domain, and two in the kinase domain [5].

Although the LRRK2 pathological mutations were first identified in 2004 [6,7], the physiological and pathological functions of LRRK2 remain enigmatic, despite the development of many cellular and animal experimental models based on LRRK2 overexpression, knock-in, or knock-out [8]. For example, the LRRK2 rodent models based on either traditional transgene or knock-in approaches poorly reflect the key features of PD in terms of dopaminergic neuron loss and motor coordination impairment. Non-rodent LRRK2 models show a more pronounced phenotype that may allow a simpler analysis of LRRK2’s pathological mechanisms or the testing of a possible therapeutic approach, albeit with all the limitations of non-high eukaryotic models, mainly in terms of neuronal architecture.

LRRK2 function has been linked to several cellular functions, although most of the studies converge on an alteration of vesicle trafficking [9,10]. In fact, to date, a subset of Rab GTPases including Rab3, Rab8, Rab10, and Rab35 are the most likely substrates of LRRK2 kinase activity [11,12]. *LRRK2* is almost ubiquitously expressed in human tissues, albeit at different expression levels in different cell types. In the brain, for example, it is expressed at high levels in the hippocampus and striatum, and at lower levels in the dopaminergic neurons in the Substantia Nigra pars compacta. *LRRK2* shows higher expression in neutrophils and myeloid cells, including monocytes and dendritic cells, suggesting an important role for LRRK2 in the interface between peripheral and central immune function in PD and, in particular, in the regulation of the innate immune system [13,14]. Indeed, *LRRK2* has been genetically linked, in humans, to different immune-related diseases—Crohn’s disease (CD) [15], leprosy [16], and systemic lupus erythematosus [17]—as well as increased susceptibility to mycobacterial infection [18]. The robust expression of LRRK2 in non-neuronal cells, coupled with the low LRRK2 expression in dopaminergic neurons, may also suggest a non-cell autonomous disease mechanism for PD associated with LRRK2 pathological mutations [19]. In this context, we have previously shown that in *Drosophila* models the LRRK2 expression in neurons (driven by Elav or nSyb promoters) does not lead to a significant pathological phenotype compared to ubiquitous expression (driven by Tubulin or Actin promoter) [20]. *Drosophila* is a valuable tool to study a possible non-neuronal LRRK2 mechanism of toxicity and, more generally, to study the state of neuroinflammation [21]. *Drosophila* has a relatively simple nervous system and a very small population of glial cells [22]. Despite the significantly lower glia/neuron ratio in flies, the sophistication of glial cell organisation in the adult fly brain still displays a very similar morphology and molecular functionality to its mammalian counterparts. Four major classes of CNS glia have been identified in *Drosophila*: cortex, neuropil, surface, and peripheral glia [22]. Cortex glia, which are also known as cell body-associated glia, are structurally very similar to astrocytes and are embedded within the cell cortex in close contact with neurons. Neuropil glia, like oligodendrocytes, are dedicated to extending sheath-like membrane structures around target axons or axon bundles. A CNS glial subtype specifically dedicated to immune functions, like mammalian microglia, does not appear to exist in the *Drosophila*; rather, all glial cells appear to be competent in performing immune-like functions such as engulfing neuronal corpses during development [22]. Interestingly, the repo-GAL4 driver results in transgene expression in all glia except the midline glia [23], making it easy to assess the overall contribution of all glia cells to the LRRK2 pathological phenotype. Interestingly, several studies of *Drosophila* show that activation of innate immunity pathways in the nervous system can contribute to neurodegeneration [21], and that inhibition of NF-κB in the glia is sufficient to delay neurodegeneration induced by proteotoxicity in neurons [24]. In *Drosophila*, the core component of the innate immune response pathways is the synthesis of powerful antimicrobial peptides (AMPs) mediated by the activation of NF-κB family transcription factors. In both *Drosophila* and mammals, AMPs tend to increase progressively with age [25], and it has been postulated that increased levels of some AMPs produced in non-neuronal tissues during aging may mediate key signals initiating neuronal aging [25].

Importantly, the LRRK2 *Drosophila* model may provide an easy way to test different compounds to determine their neuroprotective activity in the whole organism. For example, we have previously shown that levetiracetam (LEV), a compound widely used in human epilepsy treatment, is able to significantly rescue the pathological effect of LRRK2 in several high-eukaryote cell line models [26]. To date, the major LEV binding site in mammals is thought to be SV2A, an integral transmembrane protein localised to both synaptic dense core vesicles and small clear vesicles [27]. However, it has been hypothesised that LEV may have multiple targets based on its antiepileptogenic, anti-inflammatory, neuroprotective, and antioxidant properties [28]. In this regard, although SV2A is absent in invertebrates, LEV has a significant effect on *Drosophila* by regulating gene expression [29] and, more interestingly, shows neuroprotective activity in Alzheimer *Drosophila* models [30].

Therefore, based on the above considerations, by taking advantage of *Drosophila* animal models we decided to explore both the neuroinflammation state and the contribution of glial cells to neuronal toxicity in *Drosophila* expressing the LRRK2 R1441C mutant. In the present study, we demonstrate that LRRK2 R1441C expression in *Drosophila* determines an upregulation of specific AMPs already present in young animals when neurodegeneration has not yet occurred. Furthermore, glial expression of LRRK2 R1441C is significantly neurotoxic without a specific change in glia cell number. Finally, treatment with levetiracetam, a compound widely used in human therapy for epilepsy treatment, can rescue both neurodegeneration and neuroinflammation in *Drosophila* models.

## 2. Materials and Methods

### 2.1. Reagents and Solutions

Protease inhibitor cocktails (Complete Ultra tablets) were from Roche, Tween^®^ 20 (poly-ethylene glycol sorbitan monolaurate), IGEPAL^®^ CA-630 (octylphenoxy poly(ethyleneoxy)ethanol) was from Merck, Darmstadt, Germany. The phosphate-buffered saline (PBS) solution was prepared using NaCl (137 mM), KCl (2.7 mM), Na2HPO4 (8.1 mM), and KH2PO4 (1.47 mM) obtained from Merck, and then adjusted to pH 7.4. RPMI 1640 medium, Fetal Bovine Serum (FBS), and Streptomycin/Penicillin were purchased from Thermo Fischer Scientific, Waltham, MA, USA.

### 2.2. Cell Culture

Human THP-1 (ATCC^®^ TIB-202) cells were cultivated in RPMI 1640 medium with 10% FBS at 37 °C in 5% CO_2_. THP-1 cells were differentiated by incubating the cells with 1 ng/mL phorbol-12-myristate-13-acetate (PMA) (P1585 Merck, Darmstadt, Germany) for 24 h. Finally, the cells were treated for 16 h with 5 ng/mL lipopolysaccharides (LPS) (LPS25 Merck), and pretreated either with 10 µM levetiracetam (L8668 Merck) or not at all.

### 2.3. Drosophila Lines

Fly stocks were cultured on standard cornmeal medium at 25 °C with a 12 h:12 h light–dark cycle. The Actin-GAL4 driver (BDSC #4414) and Repo-GAL4 (BDSC #7415) were from Bloomington Stock Center (Bloomington, IN, USA). The UAS-LRRK2 WT and UAS-LRRK2-R1441C were a generous gift from Prof. Cheng-Ting Chien (National Taiwan University Hospital Yun-Lin Branch, Taipei, Taiwan) [31].

### 2.4. Evaluation of mRNA Expression by RT-PCR

RNA was purified from eukaryotic cells (roughly 1 × 10^6^), *Drosophila* bodies (2 male and 2 female), or *Drosophila* heads (6 male and 6 female) for each genotype in 500 µL of TRIZOL solution, according to the manufacturer’s instructions (Life Technology, Thermo Fischer Scientific, Waltham, MA, USA). In total, RNA from 1 µg of eukaryotic cells or *Drosophila* bodies and 0.5 µg of *Drosophila* heads was converted to cDNA by AMV reverse transcriptase (Promega, Madison, WI, USA) at 37 °C for 1 h. PCR amplification was performed at 94 °C for 30 s, 55 °C for 30 s, and 72 °C for 30 s using the primers indicated in [32] for *Drosophila*, and the following for human eukaryotic cells (GADP: TCATCTTCTAGGTATGACAACG and TTCCTCTTGTGCTCTTGCTG; IL-6: AGACAGCCACTCACCTCTTCAG and TTCTGCCAGTGCCTCTTTGCTG; TNF-α: AGGCGGTGCTTGTTCCTCAG and GGCTACAGGCTTGTCACTCG; CXCL10: GTGGCATTCAAGGAGTACCT and TGATGGCCTTCGATTCTGGA; IL-1β: GGCATCCAGCTACGAATCTC and GCATCTTCCTCAGCTTGTCC). Real-time PCR was performed by a Rotor-Gene Q Thermocycler and using SYBR green (Biorad, Hercules, CA, USA). The delta–delta Ct method (2^−∆∆Ct^) normalized by machine internal software was used to evaluate the differences between the samples.

### 2.5. Western Blot Analysis

Western blot analysis was performed as previously described [27]. In detail, the protein extracts were obtained by direct lysis in Laemmli buffer or NP40 1% buffer when the protein amount was quantified by a Bradford protein assay (Merck). Equal amounts of protein extracts were separated by SDS-PAGE and then electroblotted into nitrocellulose membrane (Thermo Fisher Scientific). Membranes were incubated with 3% low-fat milk in 1X PBS-Tween 0.05% solution for 1 h and then, in the same solution, with the indicated antibody: anti-LRRK2 (1:5000 MJFF2 c41-2 Epitomics, Burlingame, CA, USA), anti-phospo-RAB10 (T73) (1:1000 MJF-R21 Abcam, Cambridge, UK), anti-beta-actin (1:5000 A5441 Sigma-Aldrich, St. Louis, MO, USA), anti-alpha-tubulin (1:2000 12G10 DSHB, Iowa City, IA, USA), for 16 h at 4 °C. Goat anti-mouse immunoglobulin G (IgG) peroxi-dase-conjugated antibody (1:2500 Millipore Corporation, Burlington, MA, USA) or goat anti-rabbit IgG peroxi-dase-conjugated antibody (1:5000 Millipore Corporation) were used to reveal the immunocomplexes by enhanced chemiluminescence (ECL start, Euroclone SpA, Pero, Italy).

### 2.6. Whole-Mount Immunostaining of the Adult Drosophila Brains

Fluorescent immunostaining was performed on whole-mount dissected adult brains at 45 days of age. Cohorts of 10 flies per genotype were used each time. Brains were fixed with 4% paraformaldehyde in PBS 1X for 20 min. After washing in PBS, the brains were permeabilized by 0.3% Triton X-100 in PBS for 20 min at room temperature (RT), and then incubated in blocking buffer (5% normal goat serum in PBS 1X-0.3% Triton X-100) for 1 h at RT. Subsequently, the incubation with primary antibody anti-TH (1:500 AB152, Merck Millipore) and anti-REPO (1:50 8D12 DSHB) diluted in blocking buffer was carried out for 48 h at 4 °C. After extensive washing, the brains were incubated by secondary antibody Alexa Fluor^®^ 546 (Thermo Fisher Scientific) diluted 1:1000 in blocking solution for 48 h at 4 °C. Finally, the brains were mounted using Mowiol mounting medium, and fluorescence was revealed with a Leica TCS SP5 confocal microscope with LAS lite 170 image software. DA neurons located in the PPL1 were counted in both hemispheres after Z-stack image acquisition to visualize all TH-positive neurons. A minimum of 10 brains were analysed for each genotype in a single experiment. Three independent experiments were performed. Statistical analysis was performed with one-way ANOVA, followed by the Bonferroni post hoc test.

### 2.7. Climbing Assay

Flies of both genders were matched for age and sex, randomly selected, anesthetized by ice, and placed in conical tubes with a diameter of 2 cm. After 15 min of recovery at 25 °C, the flies were tapped to the bottom of the tube, and the climbing activity was quantified as the percentage of flies that reached 8 cm (for 7-day-old animals) or 5 cm (for 45-day-old animals) in 10 s. Any experimental sample was performed in duplicate (each with 15 flies) and the assay was repeated three/four times every 5 min.

### 2.8. Statistical Analysis

The results are shown as the means ± SEM of independent experiments, as indicated. Statistical evaluation was performed by Student’s t-test or by one-way ANOVA and Bonferroni’s multiple comparison post hoc test. Values significantly different from the relative control are indicated with one, two, or three asterisks when *p* < 0.05, *p* < 0.005, and *p* < 0.001, respectively.

## 3. Results

### 3.1. Analysis of Inflammatory Peptide Expression in LRRK2 Actin-GAL4

We have previously shown that the ubiquitous expression of LRRK2 pathological mutants in *Drosophila* determines a severe pathological phenotype compared to neuronal expression [20]. To evaluate the contribution of inflammation to this pathological phenotype, in a preliminary experiment, we used semiquantitative RT-PCR to assess the mRNA expression levels of two different AMP peptides (Attacin-A1 and Drosocin) in the whole bodies of 7- or 45-day-old flies. As shown in Figure 1A, the LRRK2 R1441C expression in both young and old flies is higher compared to that of the controls.

Interestingly, the AMP expression is also increased in the brain, as shown for AttA (Figure 1B). The data indicate that the expressions of some peptides are altered in R1441C transgenic animals as young as 7 days of age, suggesting an important inflammatory contribution to the LRRK2 phenotype in *Drosophila*, which, in our experimental model, occurs in old animals [20]. Driven by these results, we evaluated the expressions of different AMPs in the brains of 7-day-old animals through real-time PCR (Figure 1C,D). Already, at 7 days, the LRRK2 R1441C expression determines significant increases in some AMPs, both in the body and in the brain, compared to control animals.

### 3.2. Analysis of LRRK2 Phenotype when Expressed in Glial Cells

Based on previous results and the importance of non-cell autonomous pathogenic mechanisms in several neurological disorders, we generated an LRRK2 *Drosophila* model in which the LRRK2 expression is driven by a glial promoter (Repo-GAL4) compared to a ubiquitous promoter (Actin-GAL4) or control flies. Through a western blot using brain protein extracts, we confirmed the LRRK2 R1441C expression in the two different driver lines (Figure 2A). Furthermore, as previously evidenced [20,33], the LRRK2 expression in *Drosophila* strongly induces a significant increase in RAB10 phosphorylation (pRAB10) in the presence of both GAL4 drivers, further validating our experimental model (Figure 2A,B).

We then compared the locomotor activity of the different fly lines at 7 and 45 days of age. As shown in Figure 2C,D, no differences were detected at 7 days, whereas a significant decrease in climbing activity was detected in the presence of Actin and Repo drivers, suggesting a significant contribution of glial cells to the LRRK2 pathological phenotype in *Drosophila*. To further support the idea that ubiquitous or glial-specific expression of LRRK2 R1441C induces neurodegeneration, we used the decrease in the dopaminergic neuron number as a read-out. Flies were sacrificed at 45 days and whole-mount immunofluorescence was performed to analyse the Tyrosine Hydroxylase (TH)-positive neurons. As illustrated in Figure 2E,F, LRRK2 R1441C expression significantly reduces the number of TH-positive neurons in the PPL1 brain areas of both GAL4 drivers.

To investigate a possible molecular mechanism of neurodegeneration we examined, using real-time PCR, the expression levels of different anti-inflammatory peptides (AMPs) in the heads of the different genotypes. As illustrated in Figure 3A, the ubiquitous or glial expression of LRRK2 R1441C determines a significant increase in specific AMPs, suggesting an important inflammatory status in the brains of both genotypes.

Finally, to exclude the possibility that the LRRK2 R1441C expression in repo-GAL4 files could potentially induce toxicity in glial cells, we performed an immunofluorescence experiment with anti-REPO antibody on the whole brains of 45-day-old *Drosophila*. Our analysis revealed no significant differences in the numbers of glial cells between the two genotypes in the cortex area (Figure 3B).

### 3.3. Levetiracetam’s Effect on Neurodegeneration in Drosophila LRRK2 Models

As mentioned in the Introduction, we have previously demonstrated that LEV, a compound widely used in human epilepsy treatment, is able to significantly reduce the LRRK2 pathological effect in highly eukaryotic cell lines [26]. Therefore, we chronically treated our Actin-GAL4/LRRK2 R1441C *Drosophila* models with LEV. Since a previous research article has evidenced the toxicity of LEV in *Drosophila* at a dose of 7.5 mg/kg of food [30], in a preliminary experiment, we evaluated the effects of LEV at 5 or 2.5 mg/kg of food on fly vitality and climbing at 7 and 45 days after eclosion. Under our experimental conditions, an LEV dose of 5 mg/kg resulted in a slight reduction in both vitality and climbing ability; therefore, we decided to proceed with a dose of 2.5 mg/kg for all subsequent experiments.

The Actin-GAL4/LRRK2 R1441C and controls were either chronically treated with LEV or not treated at all, and evaluated after 45 days for motor activity (by climbing assay), neuronal degeneration (by dopaminergic cell analysis), and inflammatory status (by measuring the expression levels of different AMPs).

We confirmed a significant reduction in locomotor activity in *Drosophila* expressing LRRK2 R1441C, while the LEV treatment rescued the climbing defects in R1441C transgenic flies (Figure 4A).

In addition, chronic LEV treatment significantly rescued the decrease in dopaminergic neurons in *Drosophila* brains after 45 days of treatment (Figure 4B,C).

Finally, to assess the effect of LEV on neuroinflammation, we analysed Att-A1 and Drosocin expression in the brains of the different genotypes. As indicated in Figure 4D, a significant reduction in inflammatory peptides was detected in the LRRK2 R1441C flies chronically treated by LEV.

Some experimental evidence indicates a direct effect of LEV on inflammation through an unknown molecular mechanism [34,35]. To assess whether LEV treatment has a direct effect on neuroinflammation, or whether it is a secondary effect to the reduction of neurodegeneration, we investigated the effect of acute LEV treatment on human THP1 cells. Different inflammatory markers were evaluated by real-time PCR in THP1 after 16 h of LPS exposure, pretreated with LEV or not at all. No significant effects of LEV were observed in the inflammatory markers significantly induced by LPS treatment (Figure 5A). LEV’s lack of effect on THP1 cells may suggest that there is no direct effect of LEV on inflammatory cells, whereas LEV’s effect on *Drosophila* is likely to be mediated through neuronal cells, which, in turn, may modulate AMP transcriptional activation in glial cells.

## 4. Discussion

Despite promising findings, challenges remain in fully understanding the complexity of LRRK2-mediated neuronal toxicity. Elucidating the precise mechanisms of LRRK2 pathogenicity and distinguishing context-dependent effects will also be crucial for the development of a possible therapeutic approach for PD. In this context, the intersection of LRRK2 and inflammation opens a new scenario in our understanding of neurodegenerative disorders. Various experimental evidence in different neurodegenerative disease models based on mutant gene expression points to a non-cell autonomous mechanism of neuronal toxicity. In particular, the glial cells appear to play a prominent role, probably by regulating the inflammatory processes. Importantly, *LRRK2* is highly expressed in both microglia cells and astrocytes, the two main players in the regulation of immune responses to pathological processes in the brain, and seems to play a key role in the modulation of inflammation [36]. For example, various insults lead to a significant increase in inflammatory cytokine and chemokine synthesis in either microglia or astrocytes carrying the LRRK2 G2019S or R1441G mutations [37,38,39,40]. Further validating these observations, *LRRK2* knock-out (KO) [41,42] or pharmacological inhibition [43,44] show an opposite effect on the generation of inflammatory mediators. In contrast to what has been observed in brain cells, some discrepancies regarding the effect of LRRK2 on the generation of inflammatory cytokines/chemokines have been demonstrated in systemic immune cells. Taken together, LRRK2 seems to play a prominent role in immune modulation, probably in a cell type-dependent manner and depending on the specific inflammatory insult. In this context, we have used an experimental *Drosophila* model to investigate the roles of glial cells and inflammatory processes in the neuronal toxicity induced by LRRK2 R1441C expression. We were able to show that whole-organism expression of LRRK2 R1441C, driven by the actin-GAL4 driver, leads to systemic and neuronal inflammation in *Drosophila* as young as 7 days of age, when no signs of *Drosophila* pathology appear, as analysed by fly mobility. There is extensive experimental evidence that AMPs play a key role in the regulation of neuronal functions such as sleep, memory, and ageing, and more importantly, in neurodegenerative diseases. In *Drosophila*, immune cells, such as mammalian microglia, are absent; rather, all glial cells appear to be competent in performing immune-like functions. Using Repo-GAL4 drivers, we were able to show that LRRK2 R1441C expression leads to a pathological phenotype. Using dopaminergic neurons as a readout, we were able to show a significant reduction in the number of these cells. The pathological phenotype is associated with a strong and significant reduction in *Drosophila* coordination and a significant increase in mRNA-specific AMPs including AttA, Drosocin, and Metchnikowin. In *Drosophila*, numerous studies show that ageing leads to a significant increase in AMPs [45,46,47,48]. Importantly, the conserved Relish/NF-κB immune signalling pathway appears to be dysregulated in glial cells during functional ageing, leading to a substantial age-dependent increase in multiple AMPs in the brain, resulting in a reduced lifespan [46].

Remarkably, in a fly model of Alzheimer’s disease, the expression of human amyloid-β peptide (Aβ)42 caused cell death and tissue degeneration, and this phenotype was alleviated by mutations in Toll and its downstream effectors, including Dorsal and Dif [49]. Furthermore, in *Drosophila*, retinal degeneration caused by a mutation in the eye-specific phospholipase C encoded by norpA has been suppressed by mutations in Dredd and Relish, suggesting the involvement of Imd pathway components [50].

We have previously shown that levetiracetam (LEV), a drug widely used in humans for the treatment of epilepsy, is able to significantly rescue the pathological effect of LRRK2 in cell lines [26].

The molecular mechanism of action of LEV is still cryptic. Although the main target is thought to be SV2A, a neuronal protein associated with intracellular vesicles, several other targets have been hypothesised [28] mainly based on the multiple properties of LEV, which are antiepileptogenic, anti-inflammatory, neuroprotective, and antioxidant. For example, LEV attenuates inflammatory responses in microglia activated by lipopolysaccharide (LPS) [51,52].

Moreover, although SV2A is absent in invertebrates, several pieces of experimental evidence underline a significant LEV effect in *Drosophila* that regulates both gene expression [29] and neuroprotection in Alzheimer *Drosophila* models [30].

In our experimental conditions, chronic LEV treatment in *Drosophila* leads to a significant rescue of the LRRK2 R1441C pathological phenotype in terms of motility (climbing assay), neurotoxicity (number of dopaminergic neurons), and expression of inflammatory peptides (AttA and Dro). Importantly, the low doses of LEV used by us and others, as well as the absence of the SV2A protein in *Drosophila*, suggest the existence of other unidentified targets of LEV. Furthermore, LEV’s lack of effect on LPS-induced inflammation in human THP1 cells (Figure 5A) suggests that the anti-inflammatory effect of LEV in *Drosophila* may be a secondary effect of modulating neuronal activity. Further experiments are needed to identify the LEV targets that modulate the LEV anti-inflammatory effect in both *Drosophila* and mammalian systems. For example, we did not investigate any effect of LEV in microglia cells, which are the main immune cells in the brain compared to monocytes.

## 5. Conclusions

Despite all the limitations of non-mammalian models, especially in terms of neuronal architecture, *Drosophila* LRRK2 models show a more pronounced phenotype that may allow a simpler analysis of LRRK2 pathological mechanisms or the testing of a possible therapeutic approach. In this context, in our *Drosophila* models we were able to highlight a significant contribution of glial cells to LRRK2 toxicity. To our knowledge, this is the first demonstration in an animal model of the pathological effect of LRRK2 in glial cells, leading to severe inflammation and neurodegeneration, strongly supporting the hypothesis of a non-cell autonomous disease mechanism for PD associated with LRRK2 pathological mutations. In addition, we have shown that levetiracetam—a drug widely used in human epilepsy treatment—is able to reduce neuroinflammation, motor deficits, and neurodegeneration in LRRK2 *Drosophila* models. Although SV2A is considered to be a major target of LEV in mammals, other cellular targets may contribute to LEV effects, including sodium channels, calcium channels, or the GABAergic system [28], which may collectively exert the neuroprotective activity demonstrated in various experimental models.

## Figures and Tables

**Figure 1 biomedicines-12-01555-f001:**
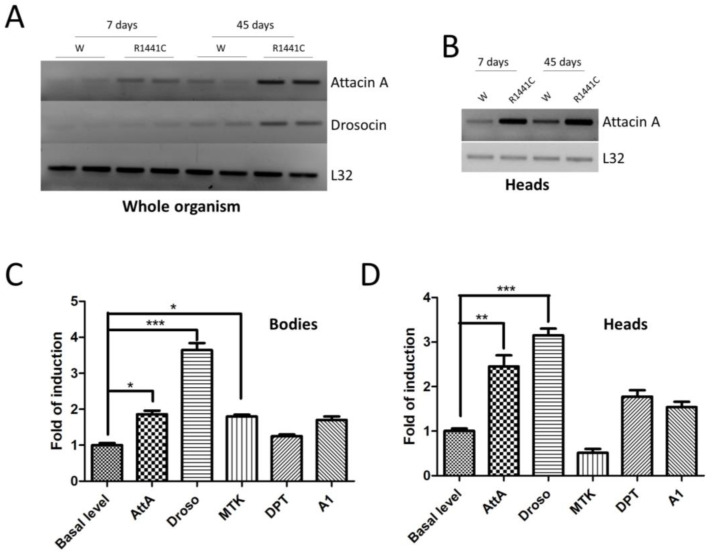
Analysis of LRRK2 R1441C expression and AMP expression. (**A**) The evaluation of AMP expression in the whole organisms of Actin-GAL4/LRRK2 R1441C and control flies. *Drosophila* of the two genotypes were sacrificed at 7 and 45 days of age, semi-quantitative RT-PCRs were performed for the indicated AMP, and the PCR products were analysed on agarose gel. L32 was used as a control to normalize the different samples. (**B**) The evaluation of AttA expression in the heads of Actin-GAL4/LRRK2 R1441C and control flies. *Drosophila* of the two genotypes were dissected at 7 and 45 days of age, semi-quantitative RT-PCRs were performed for the indicated AMP, and the PCR products were analysed on agarose gel. L32 was used as a control to normalize the different samples. (**C**,**D**) The evaluation of AMP expression by qPCR in the bodies (**C**) or heads (**D**) of the different genotypes. *Drosophila* were dissected at 7 days of age and qPCRs were performed for the indicated AMPs. L32 was used as a control to normalize the different samples. The data represent the mean ± SEM. * *p* < 0.05, ** *p* < 0.01, *** *p* < 0.001. One-way ANOVA and Bonferroni post hoc test were used.

**Figure 2 biomedicines-12-01555-f002:**
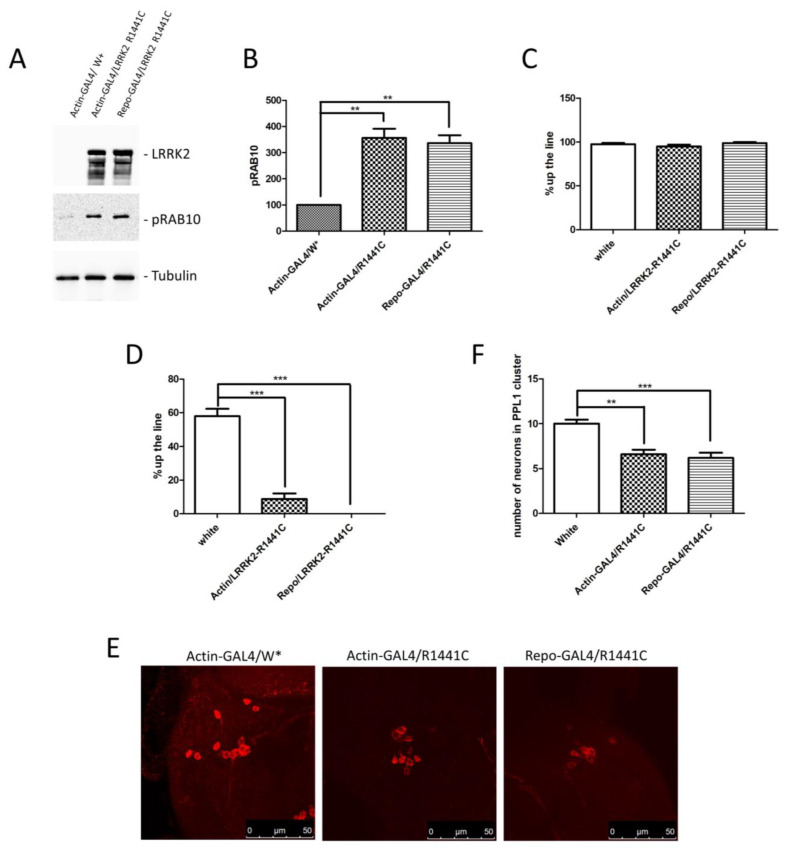
Analysis of *Drosophila* expressing LRRK2 R1441C under control of Actin- or Repo-GAL4 driver. (**A**) Evaluation of LRRK2 and pRAB10 expression in Actin- or Repo-GAL4/LRRK2 R1441C lines. One-week-old *Drosophila* from indicated genotypes were sacrificed and their heads were dissected. Protein extracts were separated by SDS-PAGE and analysed by western blot using anti-LRRK2 and anti-pRAB10. Anti-Tubulin was used as loading control. (**B**) Relative band densitometry for pRAB10, with data obtained from three independent experiments normalised to Tubulin. pRAB10 of the control is indicated as 100%. (**C**,**D**) Evaluation by climbing assay of locomotor activity of (**C**) 7- and (**D**) 45-day-old *Drosophila* lines expressing LRRK2 R1441C under the control of Actin- or Repo-GAL4 driver compared to control animals. (**E**) Dopaminergic staining of PPL1 areas of 45-day-old *Drosophila* lines of different genotypes, by immunofluorescence on whole brains using anti-TH antibody. (**F**) Quantification of dopaminergic neuronal numbers of PPL1 of different genotypes (W*: White mutant). The data represent the mean ± SEM. ** *p* < 0.01, *** *p* < 0.001. One-way ANOVA and Bonferroni post hoc test were used.

**Figure 3 biomedicines-12-01555-f003:**
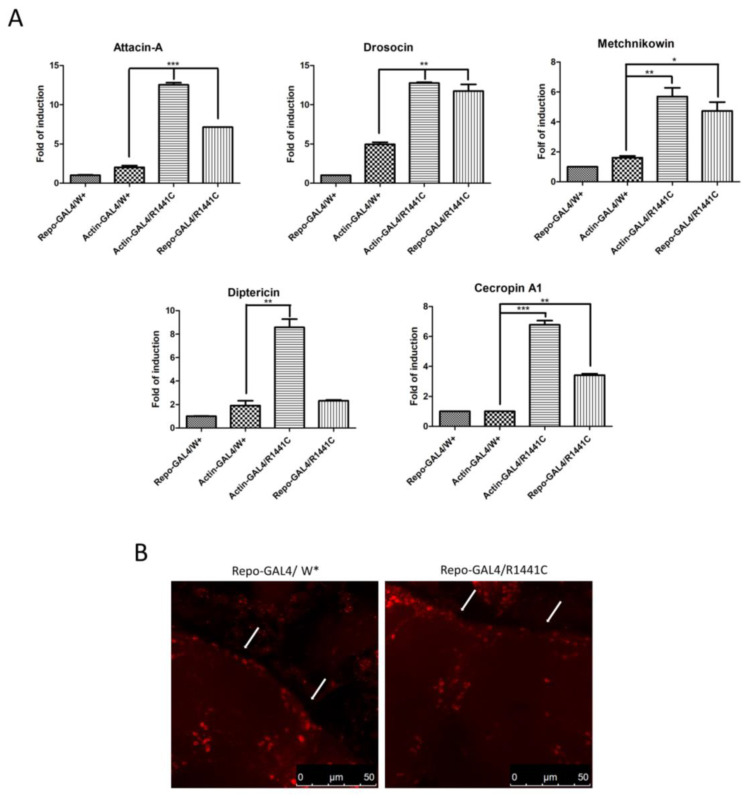
Evaluation of AMP expression through real-time PCR of heads of different genotypes and anti-REPO staining. (**A**) *Drosophila* were dissected at 45 days of age and real-time PCRs were performed for indicated AMPs. L32 was used as control to normalize different samples. Data represent mean ± SEM. * *p* < 0.05, ** *p* < 0.01, *** *p* < 0.001. One-way ANOVA and Bonferroni post hoc test were used. (**B**) Anti-REPO staining on whole brains of 45-day-old *Drosophila* of two different genotypes by immunofluorescence. The white arrows indicate groups of REPO-positive cells; W*: White mutant.

**Figure 4 biomedicines-12-01555-f004:**
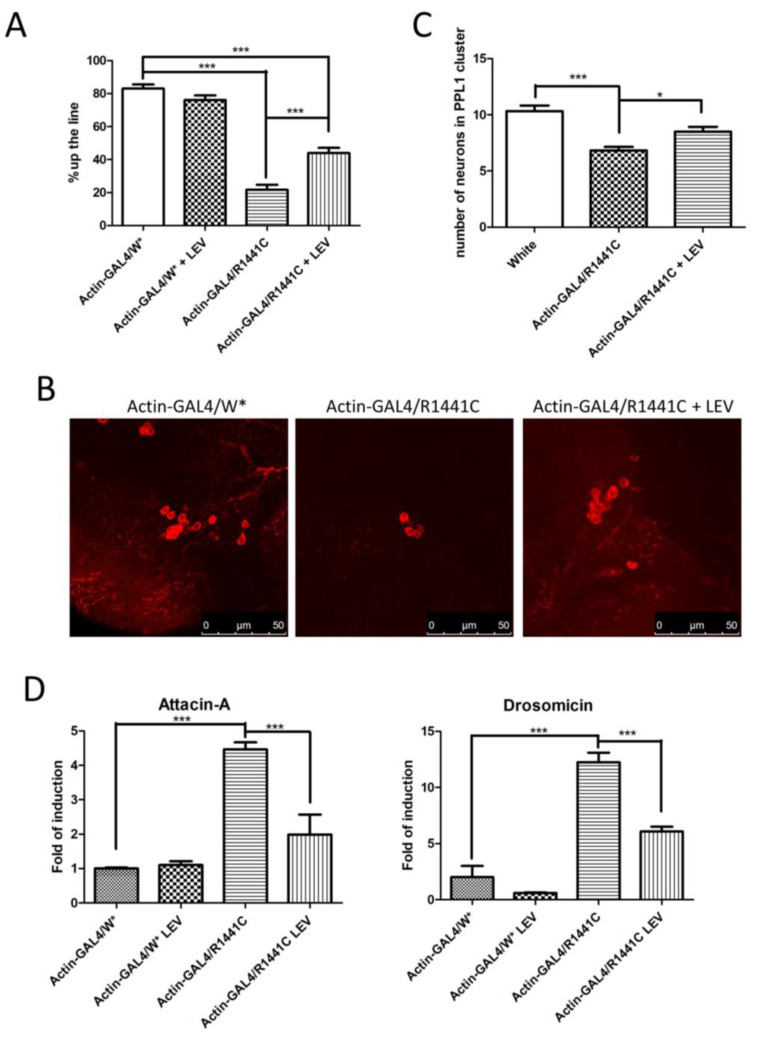
Analysis of *Drosophila* expressing LRRK2 R1441C under control of Actin-GAL4, chronically treated by LEV for 45 days or not treated at all. (**A**) Evaluation, by climbing assay, of locomotor activity of 45-day-old *Drosophila* lines expressing LRRK2 R1441C under control of Actin-GAL4 driver, compared to control animals, upon LEV treatment. (**B**) Dopaminergic staining of PPL1 areas of 45-day-old *Drosophila* lines of different genotypes by immunofluorescence on whole brains using anti-TH antibody. (**C**) Quantification of dopaminergic neuronal numbers of PPL1 of different genotypes. (**D**) Evaluation of AMP expression by real-time PCR in heads of different genotypes. *Drosophila* were dissected at 45 days of age and real-time PCRs were performed for indicated AMPs. L32 was used as control to normalize different samples. W*: White mutant. The data represent the mean ± SEM. * *p* < 0.05, *** *p* < 0.001. One-way ANOVA and Bonferroni post hoc test were used.

**Figure 5 biomedicines-12-01555-f005:**
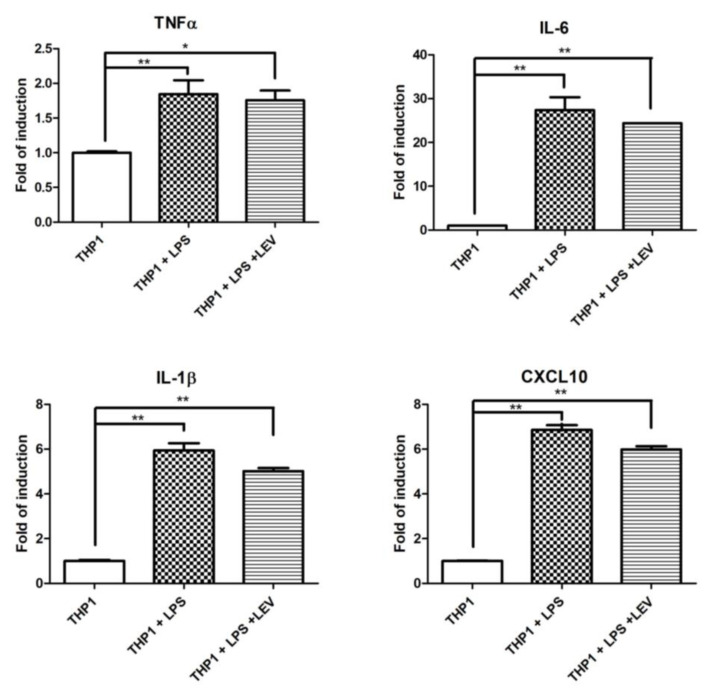
Analysis of LEV’s direct effect on inflammation in THP1 cells and *Drosophila*. qPCR evaluation of TNF-α, IL-1β, IL-6, and CXCL10 mRNA expression levels in THP1 cells upon 16 h of LPS treatment, either pretreated by LEV or not at all. The data represent the mean ± SEM. * *p* < 0.05, ** *p* < 0.01. One-way ANOVA and Bonferroni post hoc test were used.

## Data Availability

The data underlying the findings presented in this study are available upon reasonable request.

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
