# Peer review of "Evaluation of Neuroinflammatory Contribution to Neurodegeneration in LRRK2 Drosophila Models"

_biomedicines, 2024, doi:10.3390/biomedicines12071555_

Round 1

Reviewer 1 Report

Comments and Suggestions for Authors

The present work entitled “Evaluation of neuroinflammatory contribution to neurodegeneration in LRRK2 drosophila models” is a very interesting study that shows the effect of LRRK2 on inflammatory compounds and neurodegeneration in the drosophila model. However, the study shows important deficiencies that must be resolved.

First, the authors do not describe the main compound of the study, Levetiracetam, neither in the introduction (as indicated in the results section, page 8, line 293) nor in the results. The study lacks a description of the type of compound, biological actions, therapeutic targets... and an explanation of why it affects invertebrates when they do not have SV2A, and why it is also used in other neurological disorders different from epilepsy.

There is no information about the cells used in the last figure, THP1 cells. It is not known what type of cells they are, from what animal, how they are cultured, etc. There is also no information about the treatments in these cells, whether LPS or LEV, the doses, time... since it will be different from those of animals. Instead, other reagents appear that do not seem to be used in the experiments, such as the LRRK2 inhibitors: CZC-25146 130 (Calbiochem).

One of the most repeated results in the study is the counting of TH neurons as the only measure of neurodegeneration, however, it is not explained how this counting is carried out, which makes the study non-reproducible.

Furthermore, another weakness of the study is the non-quantification of protein expression. The protein study is limited to a single type of experiment and only qualitative, which is not understandable having access to animal material.

In summary, the study is very interesting, but it is incomplete, especially in establishing the role of LEV in the process. Additional experiments are needed.

Comments on the Quality of English Language

Minor editing of English language required

Author Response

Dear Reviewer 1,

Please find attached a revised version of our manuscript entitled "Evaluation of neuroinflammatory contribution to neurodegeneration in LRRK2 drosophila models". We would like to thank both reviewers for their relevant comments and valuable suggestions to improve the quality of our manuscript. We have addressed all the issues they raised point by point (see below). We hope that this new version will be suitable for publication in Biomedicines.

Point by point answer:

The present work entitled “Evaluation of neuroinflammatory contribution to neurodegeneration in LRRK2 drosophila models” is a very interesting study that shows the effect of LRRK2 on inflammatory compounds and neurodegeneration in the drosophila model. However, the study shows important deficiencies that must be resolved.

First, the authors do not describe the main compound of the study, Levetiracetam, neither in the introduction (as indicated in the results section, page 8, line 293) nor in the results. The study lacks a description of the type of compound, biological actions, therapeutic targets... and an explanation of why it affects invertebrates when they do not have SV2A, and why it is also used in other neurological disorders different from epilepsy.

We apologise for the lack of description of levetiracetam in the Introduction. We have added this new section, which is highlighted in yellow.  Please note that we have shortened the introduction and added a conclusion as requested by reviewer2.

There is no information about the cells used in the last figure, THP1 cells. It is not known what type of cells they are, from what animal, how they are cultured, etc. There is also no information about the treatments in these cells, whether LPS or LEV, the doses, time... since it will be different from those of animals. Instead, other reagents appear that do not seem to be used in the experiments, such as the LRRK2 inhibitors: CZC-25146 130 (Calbiochem).

Thanks to the referee for pointing this out. We have added a specific paragraph in the MM section entitled "Cell culture". In general, THP1 is a monocyte-like line isolated from the peripheral blood of a patient with acute monocytic leukaemia that can be readily differentiated to the macrophage phenotype by phorbol-12-myristate-13-acetate (PMA). In addition, THP1 responds rapidly to several inflammatory molecules, including lipopolysaccharide (LPS). They are widely used to evaluate inflammation and anti-inflammatory molecules in various experimental assays.

We apologize for the presence of LRRK2 inhibitor description, that is a text refuse.

One of the most repeated results in the study is the counting of TH neurons as the only measure of neurodegeneration, however, it is not explained how this counting is carried out, which makes the study non-reproducible.

The following sentence has been added to the Materials and methods section in "Whole mount immunostaining of adult Drosophila brain" paragraph: "DA neurons, located in the PPL1 region, were counted in both hemispheres after Z-stack image acquisition to visualize all TH-positive neurons. A minimum of 10 brains per genotype were analysed in each experiment. Three independent experiments were performed. Statistical analysis was performed using one-way ANOVA followed by Bonferroni post hoc test”

Furthermore, another weakness of the study is the non-quantification of protein expression. The protein study is limited to a single type of experiment and only qualitative, which is not understandable having access to animal material.

All experiments were performed at least three times. We have added protein quantification (Fig. 2B) from three different experiments. To normalise the different experiments, the control pRAB10 is shown as 100%.

We hope that the new version has clarified the various issues that have been raised.

Reviewer 2 Report

Comments and Suggestions for Authors

The authors presented an interesting article. I have several suggestions for this article.

1) In my opinion, the introduction is too redundant and I would recommend shortening it

2) Immunostaining photographs for Figures 2 and 4 should be enlarged

3) The authors should write a separate conclusion and, in conclusion, indicate the advantages and limitations of the research model used.

Author Response

Dear Reviewer 2,

Please find attached a revised version of our manuscript entitled "Evaluation of neuroinflammatory contribution to neurodegeneration in LRRK2 drosophila models". We would like to thank both reviewers for their relevant comments and valuable suggestions to improve the quality of our manuscript. We have addressed all the issues they raised point by point (see below). We hope that this new version will be suitable for publication in Biomedicines.

Point by point answer:

The authors presented an interesting article. I have several suggestions for this article.

In my opinion, the introduction is too redundant and I would recommend shortening it

We reduced the introduction as required although we expanded the Levetiracetam description as required by referee 1.

2) Immunostaining photographs for Figures 2 and 4 should be enlarged

We enlarged the figures 2 and 4

3) The authors should write a separate conclusion and, in conclusion, indicate the advantages and limitations of the research model used.

We added a conclusion paragraph addressing the raised point.

We hope that the new version has clarified the various issues that have been raised.

Round 2

Reviewer 1 Report

Comments and Suggestions for Authors

The changes introduced in the revised version of the article substantially improved it.

Reviewer 2 Report

Comments and Suggestions for Authors

The article can be accepted for publication in its current form